# Root cap cell corpse clearance limits microbial colonization in *Arabidopsis thaliana*

**Nyasha Charura[1†], Ernesto Llamas[1†], Concetta De Quattro[1], David Vilchez[2,3,4], Moritz K Nowack[5,6‡], Alga Zuccaro[1*‡]**

[1]Cluster of Excellence on Plant Sciences (CEPLAS), Institute for Plant Sciences, University of Cologne, Cologne, Germany; [2]Cologne Excellence Cluster for Cellular Stress Responses in Aging-Associated Diseases (CECAD), University of Cologne, Cologne, Germany; [3]Center for Molecular Medicine Cologne (CMMC), University of Cologne, Cologne, Germany; [4]Faculty of Medicine, University Hospital Cologne, Cologne, Germany; [5]Department of Plant Biotechnology and Bioinformatics, Ghent University, Ghent, Belgium; [6]VIB Center for Plant Systems Biology, Ghent, Belgium

**\*For correspondence:**
azuccaro@uni-koeln.de

[†]These authors contributed equally to this work
[‡]These authors also contributed equally to this work

**Competing interest:** The authors declare that no competing interests exist.

## eLife assessment

This study investigated the involvement of programmed cell death (PCD) in *Arabidopsis thaliana* root cap cells and its effect on microbial colonization. The authors have reported the importance of timely corpse clearance in the root cap and a root cap-specific transcription factor in controlling microbial colonization by beneficial fungi. By demonstrating the connection between transcriptional control of PCD and microbial colonization, this study provides **fundamental** insights into how relationships are established and regulated at the root-microbiome interface. The strength of the evidence presented is **convincing**, providing a foundation for further research concerning the spatial and temporal dynamics of microbiome recruitment along the root axis.

**Abstract** Programmed cell death occurring during plant development (dPCD) is a fundamental process integral for plant growth and reproduction. Here, we investigate the connection between developmentally controlled PCD and fungal accommodation in *Arabidopsis thaliana* roots, focusing on the root cap-specific transcription factor ANAC033/SOMBRERO (SMB) and the senescence-associated nuclease BFN1. Mutations of both dPCD regulators increase colonization by the beneficial fungus *Serendipita indica*, primarily in the differentiation zone. *smb-3* mutants additionally exhibit hypercolonization around the meristematic zone and a delay of *S. indica*-induced root-growth promotion. This demonstrates that root cap dPCD and rapid post-mortem clearance of cellular corpses represent a physical defense mechanism restricting microbial invasion of the root. Additionally, reporter lines and transcriptional analysis revealed that *BFN1* expression is downregulated during *S. indica* colonization in mature root epidermal cells, suggesting a transcriptional control mechanism that facilitates the accommodation of beneficial microbes in the roots.

## Introduction

Plant roots elongate by producing new cells in the root apical meristem at the root tip. As roots extend further into the soil environment, the newly formed root tissue is naturally exposed to microbial attack, challenging the successful establishment of root systems. However, microbial colonization

**eLife digest** As plant roots grow deeper into the soil, they encounter various fungi and bacteria. Some of these microbes attempt to infect the roots, but certain interactions can be mutually beneficial, promoting microbe survival while protecting plants from harmful infections. However, microbes rarely invade the root tip, which is protected by a special type of tissue called the root cap.

As roots grow, root cap cells are constantly removed and replaced. In the model plant *Arabidopsis thaliana*, root cap turnover occurs via a combination of cellular shedding and developmental programmed cell death (dPCD). Proteins known as SMB and BFN1 regulate this process by triggering cell death and ensuring dead cells are removed.

To investigate whether the rate of dPCD and the degree of post-mortem corpse clearance affect how fungi accumulate in plant roots, Charura et al. studied *Arabidopsis* plants with a non-functional SMB protein. Staining techniques revealed an accumulation of dead cells remaining in the root cap, as well as increased growth of the fungus *Serendipita indica* in the root tip. These changes also disrupted the growth-promoting effects typically initiated by the fungus. Taken together, the findings suggest that under normal conditions, SMB drives the continuous clearance of cells through dPCD, which limits fungal growth in the root tip that could otherwise harm the plant.

Charura et al. next looked at how *S. indica* infection affects the expression of genes that drive dPCD. This revealed reduced expression of the gene for BFN1 in *Arabidopsis* plants infected with the fungus. Staining the roots of plants containing a non-functional form of BFN1 also revealed increased dead cell remnants and greater fungal growth further up the root, suggesting that *S. indica* may exploit host cell clearance pathways to colonize the roots.

In conclusion, the findings show that the rate of dPCD in plant roots is key to limiting fungal invasion. The decreased BFN1 gene expression observed with *S. indica* infection suggests that fungi may manipulate BFN1 to help them form more beneficial partnerships. Understanding the interplay between root cap turnover and fungal invasion could lead to more sustainable agricultural practices and may help researchers to improve plant nutrition and tolerance without relying on chemical fertilizers or pesticides.

at the meristematic zone is rarely detected (*Jacobs et al., 2011*; *Deshmukh et al., 2006*). The sensitive tissue of meristematic stem cells is surrounded by the root cap, a specialized root organ that orchestrates root architecture, directs root growth based on gravitropism and hydrotropism, and senses environmental stimuli. In addition, the root cap is presumed to have a protective function in soil exploration (*Kumpf and Nowack, 2015*; *Moriwaki et al., 2013*).

In *Arabidopsis thaliana* (hereafter *Arabidopsis*), the root cap consists of two distinct tissues: the centrally located columella root cap at the very root tip and the peripherally located lateral root cap (LRC), which flanks both the columella and the entire root meristem (*Dolan et al., 1993*). A ring of specific stem cells continuously generates both new LRC cells and root epidermal cells (*Dolan et al., 1993*). However, despite the constant production of LRC cells, the root cap itself does not grow in size but matches the size of the meristem (*Kumpf and Nowack, 2015*; *Fendrych et al., 2014*; *Barlow, 2002*). To maintain size homeostasis, root cap development is a highly regulated process that varies in different plant species. In *Arabidopsis*, a combination of dPCD and shedding of old cells into the rhizosphere has been described (*Kumpf and Nowack, 2015*; *Bennett et al., 2010*). The centrally located columella root cap along with adjacent proximal LRC cells is shed as a cell package, followed by a PCD process (*Feng et al., 2022*; *Shi et al., 2018*; *Huysmans et al., 2018*). In contrast, LRC cells at the distal end of the root tip elongate and reach the edge of the meristematic zone where they undergo dPCD followed by corpse clearance on the root surface, orchestrated as part of a terminal differentiation program by the root cap-specific transcription factor ANAC33/SOMBRERO (SMB; *Fendrych et al., 2014*; *Bennett et al., 2010*; *Willemsen et al., 2008*). SMB belongs to a plant-specific family of transcription factors carrying a NAC domain (NAM - no apical meristem; ATAF1 and –2, and CUC2 - cup-shaped cotyledon). SMB promotes the expression of genes associated with the initiation and execution of LRC cell death, including the senescence-associated bifunctional nuclease *BFN1* and the putative aspartic protease *PASPA3* (*Huysmans et al., 2018*; *Fendrych et al., 2014*). BFN1 localizes in the ER, but upon cell death the protein is released and its nuclease activity ensures rapid

and irreversible degradation of RNA and DNA in the nucleus and cytoplasm as part of a rapid cell-autonomous corpse clearance at the root surface (*Reza et al., 2018*; *Fendrych et al., 2014*; *Farage-Barhom et al., 2011*). Accordingly, DNA and RNA degradation in *bfn1-1* loss-of-function mutants is delayed (*Fendrych et al., 2014*). Precise timing of cell death and elimination of LRC cells before they fully enter the elongation zone is essential for maintaining root cap size and optimal root growth (*Fendrych et al., 2014*). Loss of SMB activity results in a delayed cell death, causing LRC cells to enter the elongation zone where they eventually die without expression of dPCD executor genes in the root cap (*Fendrych et al., 2014*). Interestingly, the aberrant cell death of LRC cells in the elongation zone of *smb-3* mutants is not followed by corpse clearance, resulting in an accumulation of uncleared cell corpses along the entire root surface (*Fendrych et al., 2014*).

Despite its importance in root morphology and plant development, little is known about the importance of dPCD and rapid cell corpse clearance on plant-microbe interactions. To address this question, we tested two well-characterized loss-of-function T-DNA insertion lines, *smb-3* and *bfn1-1*, during colonization with *Serendipita indica*, a beneficial fungus of the order Sebacinales. As a root endophyte, *S. indica* colonizes the epidermal and cortex layers of a broad range of different plant hosts, conferring various beneficial effects, including plant growth promotion, protection against pathogenic microbes and increased tolerance to abiotic stresses (*Boorboori and Zhang, 2022*; *Mahdi et al., 2022*; *Fesel and Zuccaro, 2016*). The colonization strategy of *S. indica* comprises an initial biotrophic interaction, followed by a growth phase associated with a restricted host cell death that does not, however, diminish the beneficial effects on the plant host. The induction of restricted cell death in the epidermal and cortex layers is a crucial component of the colonization strategy of *S. indica* and is accompanied by an increased production of fungal hydrolytic enzymes (*Zuccaro et al., 2011*; *Deshmukh et al., 2006*). The switch between the biotrophic and the cell death-associated phase can vary depending on the host system and environmental conditions, but has been postulated to occur approximately 6–8 days post inoculation (dpi) in *Arabidopsis* (*Zuccaro et al., 2011*). Although several effector proteins involved in fungal accommodation have been described (*Dunken et al., 2022*; *Nostadt et al., 2020*; *Nizam et al., 2019*; *Weiß et al., 2016*; *Wawra et al., 2016*; *Akum et al., 2015*), the exact mechanism by which *S. indica* manipulates host cell death and the role of dPCD in fungal accommodation in the roots are largely unclear.

Here, we show that the accumulation of uncleared LRC cell corpses on the roots of *smb-3* mutants triggers hypercolonization by *S. indica*, especially around the meristematic zone, and delays *S. indica*-induced root growth promotion. We propose that a tight regulation of host dPCD and rapid and complete clearance of root cap cell corpses play important roles in restricting fungal colonization at the root apical meristem. Furthermore, we show that *S. indica* downregulates *BFN1* in older and differentiated epidermal cells to promote fungal accommodation. Our results emphasize that beneficial microbes have the ability to modify plant dPCD processes to enhance host colonization.

## Results
### The SMB-mediated clearance of dead cells protects the root meristem and regulates symbiosis

dPCD and corpse clearance are the final steps in LRC differentiation, maintaining root cap organ size in *Arabidopsis* root tips. This process is orchestrated by the LRC-specific transcription factor SMB and executed by its direct and indirect downstream targets (*Figure 1A*). To characterize the role of disrupted dPCD in *Arabidopsis* LRCs, we analyzed the phenotypic implications of the SMB loss-of-function allele *smb-3* (*Willemsen et al., 2008*). We employed two staining methods to visualize the extent of cell death and protein aggregation in the *smb-3* T-DNA insertion line. We first used Evans blue staining, a viability dye that penetrates damaged/dying cells (*Vijayaraghavareddy et al., 2017*). In *smb-3* mutants, this staining revealed uncleared LRC cell corpses along the surface of primary roots, starting at the distal border of the meristematic zone (*Figure 1B and C* and *Figure 1—figure supplement 1*). We further characterized *smb-3* mutants with Proteostat staining, a fluorescent dye that binds to quaternary protein structures typically found in misfolded and/or aggregated and condensed proteins (hereafter referred to as protein aggregates; *Llamas et al., 2021*). In *smb-3* mutants, Proteostat staining showed an accumulation of protein aggregates in uncleared dead LRC cells attached to the roots (*Figure 1D–F*).

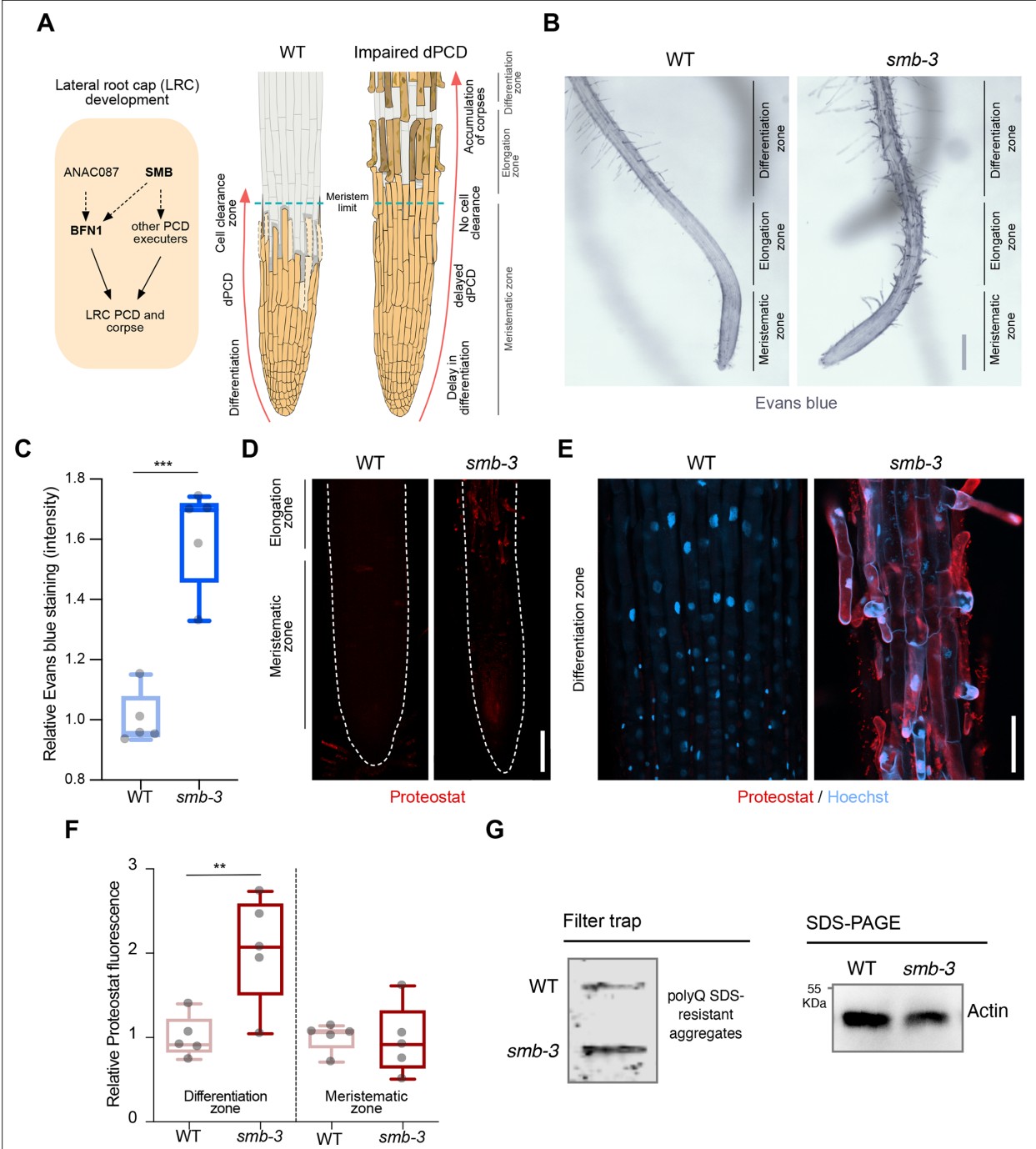

**Figure 1.** *smb-3* mutant roots exhibit uncleared cell corpses loaded with misfolded / aggregated proteins. (**A**) Schematic representation of lateral root cap (LRC) development in WT and *smb-3* mutant plants, impaired in dPCD. (**B**) Evans blue staining of 10-day-old WT and *smb-3* mutant roots, showing an overview of the meristematic, elongation and differentiation zone. Evans blue highlights the accumulation of LRC cell corpses on *smb-3* mutant roots starting at the transition into the elongation zone. Scale indicates 150 µm. (**C**) Relative quantification of Evans blue staining of the differentiation zone of 14-day-old WT and *smb-3* mutant roots (refers to image shown in B). Five plants per genotype were used. All data points were normalized to the mean of the WT control. Statistical relevance was determined by unpaired, two-tailed Student's t test (F [4, 4]=3.596; p<0.001). (**D**) Confocal Scanning Laser Microscopy (CLSM) images of 10-day-old WT and *smb-3* mutant roots stained with Proteostat (red) showing the meristematic- and the beginnings of the elongation-zone. Scale indicates 100 µm. (**E**) Magnification of the differentiation zone of WT and *smb-3* mutant roots, both stained with Proteostat (red) and Hoechst (blue). WT roots do not show any Proteostat signal, while *smb-3* mutants display extensive protein aggregate accumulation in uncleared LRC cell corpses. Scale indicates 50 µm. (**F**) Quantification of relative Proteostat fluorescence levels, comparing the differentiation and meristematic zones of WT and *smb-3* mutants. Five 10-day-old plants were used for each genotype. All data points were normalized to the mean of the WT control,

*Figure 1 continued on next page*

*Figure 1 continued*

analyzing differentiation and meristematic zone separately. Statistical significance was determined by one-way ANOVA and Tukey's post hoc test (F [3, 16]=8,314; p<0.01). (**G**) Filter trap and SDS-PAGE analysis with anti-poly-glutamine (polyQ) antibodies of 15-day-old WT and *smb-3* mutant roots. The images are representative of two independent experiments.

The online version of this article includes the following source data and figure supplement(s) for figure 1:

**Source data 1.** PDF file containing the original blots for *Figure 1G*, indicating the relevant bands.

**Source data 2.** Original files for the blots displayed in *Figure 1G*.

**Figure supplement 1.** Accumulation of LRC cell corpses on *smb-3* mutants.

The transcription factor SMB promotes the expression of various putative dPCD executor genes, including proteases that break down and clear cellular debris and protein aggregates following cell death induction. In the LRCs of *smb-3* mutants, the absence of induction of these proteases potentially explains the accumulation of protein aggregates in uncleared dead LRC cells.

Filter trap analysis further confirmed elevated levels of protein aggregates in *smb-3* mutants compared to WT roots (*Figure 1G*). Under physiological conditions in WT roots, we previously observed protein aggregate accumulation in sloughed columella cell packages, but not during dPCD of distal LRC clearance (*Llamas et al., 2021*). This aligns with the findings that dPCD of the columella is affected by the loss of autophagy, while dPCD of the LRC is not (*Feng et al., 2022*).

In *smb-3* mutants, neither Proteostat nor Evans blue staining was detected in LRC cells covering the meristem or in epidermal cells beneath the uncleared LRC cell corpses along the elongation and differentiation zone (*Figure 1B, D and F*). This observation highlights that the loss of SMB activity specifically affects the induction of dPCD in LRC cells at the transition between meristematic and elongation zone. These data show that the delayed cell death of LRC cells in *smb-3* mutants in the elongation zone is accompanied by impaired protein homeostasis (proteostasis), resulting in the accumulation of misfolded and aggregated proteins in uncleared LRC cell corpses.

To assess the effects of impaired dPCD processes in the LRC of *smb-3* mutants on plant-microbe interactions, we measured colonization rates of *S. indica*. We quantified extraradical colonization using the chitin-binding fluorescent marker Alexa Fluor 488 conjugated to Wheat Germ Agglutinin (WGA-AF 488) as a proxy for fungal biomass. We compared staining intensities of WGA-AF 488 between *S. indica*-colonized WT and *smb-3* roots. *S. indica* showed a clear hypercolonization phenotype along the main root axis of *smb-3* mutants (*Figure 2A and B*). On WT roots, *S. indica* preferentially colonized the differentiation zone, leaving the meristematic and elongation zone largely uncolonized, whereas mycelial growth was clearly detectable at the root tips of *smb-3* plants (*Figure 2A* and *Figure 1—figure supplement 1*; *Jacobs et al., 2011*; *Deshmukh et al., 2006*). Intraradical colonization by *S. indica* was quantified by comparing fungal and plant single-copy housekeeping marker genes using quantitative PCR (qPCR), after washing of roots to remove outer fungal mycelium. The results showed a significant increase in intraradical fungal accommodation in *smb-3* mutants compared to WT roots (*Figure 2C*). To assess the biological implications of hypercolonization, we measured *S. indica*-induced root growth in WT and *smb-3* mutants. While *S. indica* consistently and significantly increased root length in WT plants at 8, 10 and 14 dpi, increased length of *smb-3* mutant roots was only observable at later stages of colonization, indicating a delayed growth promotion phenotype (*Figure 2D*). Detailed cytological analysis confirmed that *S. indica* grew extensively around the meristematic zone of *smb-3* mutants but not of WT roots (*Figure 3A and B*) and showed increased colonization of *smb-3* mutants in the differentiation zone (*Figure 3C*). We further observed that *S. indica* was accommodated in cells that were subject to cell death and protein aggregation in the *smb-3* background (*Figure 3B, D and E*). Together, these findings indicate that loss of SMB activity in the root cap results in an accumulation of uncleared LRC cell corpses that promotes fungal colonization from the meristematic to the differentiation zone. Therefore, we postulate that the continuous clearance of root cap cells in WT roots is important to limit microbial colonization along the entire root axis and prevent microbial colonization in the meristematic zone.

Interestingly, Evans blue cell death staining of *S. indica*-inoculated *smb-3* mutants displayed a clearing of LRC cell corpses from the surface of *smb-3* mutant roots over time, while mock-treated *smb-3* mutant roots remained littered with LRC cell corpses (*Figure 3F* and *Figure 1—figure supplement 1*).

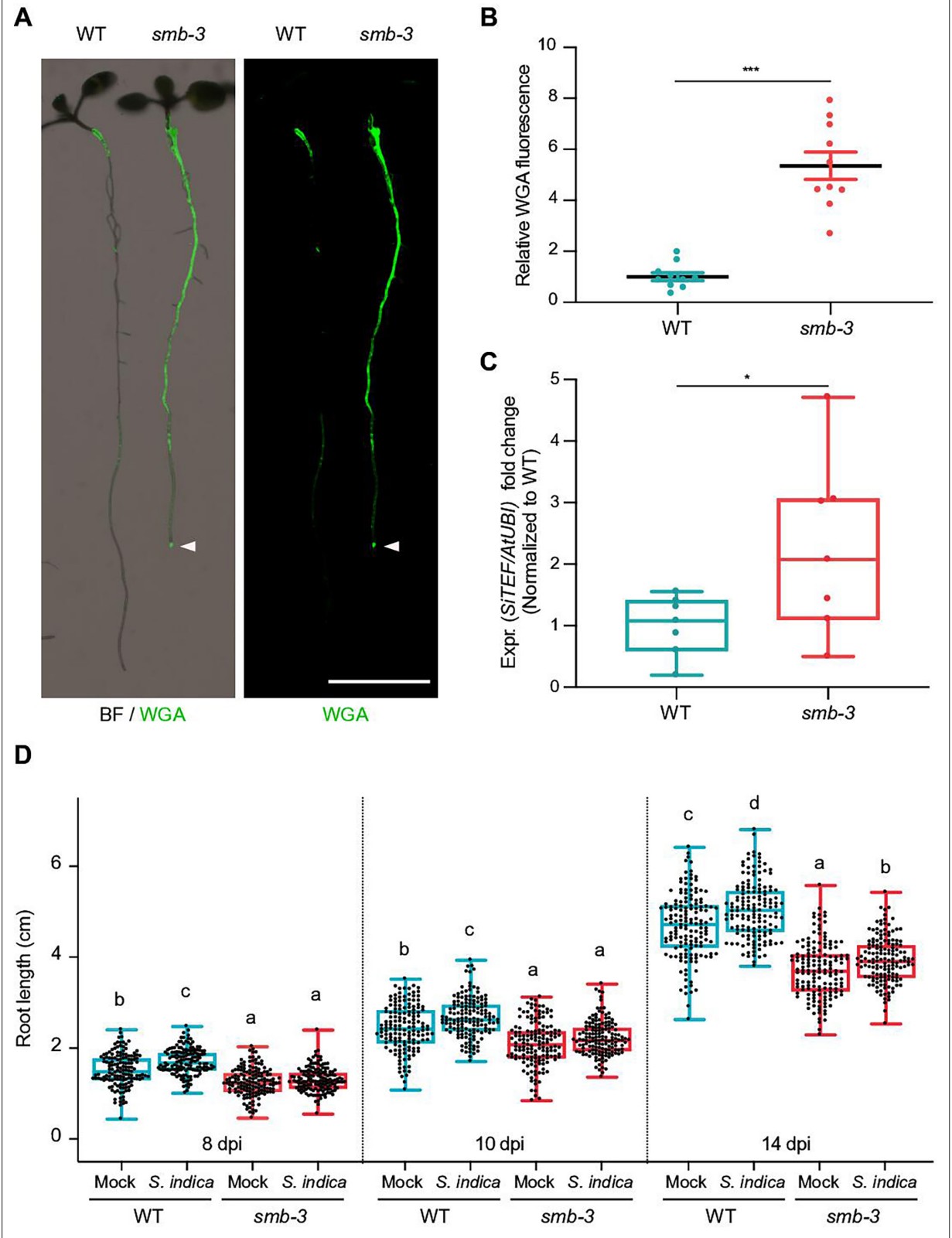

**Figure 2.** *smb-3* mutants display extraradical hypercolonization and increased intraradical colonization by *S. indica*. (**A**) Representative images show extraradical colonization of 10-day-old WT and *smb-3* mutant seedlings (seed inoculated). *S. indica* was stained with WGA-AF 488. Roots were scanned and captured with a LI-COR Odyssey M imager using the bright field (BF) and Alexa Fluor 488 channel (green). White arrowheads indicate colonization of the root tip in the *smb-3* mutant background. Scale indicates 5 mm. (**B**) Relative quantification of WGA-AF 488 signal as proxy for

*Figure 2 continued on next page*

*Figure 2 continued*

extraradical colonization on *smb-3* mutants and WT roots (refers to image shown in A). The statistical comparison was made by two-tailed Student's t test for unpaired samples (F [9, 9]=11.85; p<0.001) using 10 plants. (C) Measurement of intraradical colonization in WT and *smb-3* mutant roots at 10dpi were performed by qPCR. Roots from seven biological replicates were collected and washed to remove extra-radical hyphae, pooling approximately 30 seedlings for each genotype per replicate. The graph is normalized to WT. Statistical analysis was done via two-tailed Student's t test for unpaired samples (F [6, 6]=8.905; p<0.05). (D) Root length measurements of WT plants and *smb-3* mutant plants, during *S. indica* colonization (seed inoculated) or mock treatment. 50 plants for each genotype and treatment combination were observed and individually measured over a time period of two weeks. WT roots show *S. indica*-induced growth promotion, while growth promotion of *smb-3* mutants was delayed and only observed at later stages of colonization. This experiment was performed three times, with fresh fungal material, showing similar results. Statistical analysis was performed via one-way ANOVA and Tukey's post hoc test (F [11, 1785]=1149; p<0.001). For visual representation of statistical relevance each time point was additionally evaluated via one-way ANOVA and Tukey's post hoc test at 8 dpi (F [3, 593]=69.24; p<0.001), 10dpi (F [3, 596]=47.59; p<0.001) and 14dpi (F [3, 596]=154.3; p<0.001).

This observation indicates that *S. indica* is able to degrade uncleared cell corpses, which likely provide additional nutrients that fuel fungal hypercolonization in the *smb-3* mutant background.

## The senescence associated plant nuclease BFN1 is exploited by beneficial microbes to facilitate root accommodation

To further explore the role of root dPCD during *S. indica* accommodation in *Arabidopsis*, we performed transcriptome analysis, tracking developmental cell death-marker gene expression during different colonization stages (*Olvera-Carrillo et al., 2015*). The major regulator in LRCs, *SMB*, showed no significant changes in expression during fungal colonization (*Figure 4A and C*). However, in *Arabidopsis* colonized by *S. indica*, there was a significant decrease in *BFN1* expression observed after 6 dpi (*Figure 4B and C*). To validate the RNA-Seq analysis, we performed whole-root qPCR of WT mock- and colonized-roots, confirming *BFN1* downregulation at the onset of cell death in *S. indica*-colonized plants (*Figure 4D*).

While *SMB* expression is restricted to the LRC, *BFN1* exhibits a broader expression pattern across various cell types and tissues, such as root cap cells, cells adjacent to emerging lateral root primordia, differentiating xylem tracheary elements, as well as senescent leaves, and abscission zones of flowers and seeds (*Escamez et al., 2020*; *Farage-Barhom et al., 2008*). This widespread expression establishes BFN1 as a key player in the general regulation of dPCD and senescence processes in various tissues in *Arabidopsis*. To visualize the extent of *BFN1* downregulation upon *S. indica* colonization in different zones of the root, we used a transgenic *BFN1* promoter-reporter line (*pBFN1::NLS-tdTOMATO*) (*Huysmans et al., 2018*). In agreement with the previously described GUS reporter lines (*Farage-Barhom et al., 2008*), we detected activation of the *BFN1* promoter via accumulation of the fluorescent tdTOMATO signal in the nuclei of root cap and xylem cells in mock-treated roots (*Figure 4—figure supplement 1*). Additionally, we observed promoter activation in epidermal root cells of the differentiation zone (*Figure 4E*). In the distal region of the differentiation zone in young epidermal cells, the tdTOMATO signal was observed in nuclei, while in the basal region of the differentiation zone in older epidermal cells, the tdTOMATO signal appeared to be dispersed (*Figure 4E*, *Figure 4—figure supplement 1*). These findings indicate ongoing nuclear envelope breakdown as a hallmark of cell death (*Wang et al., 2024*) in the older part of the root, independent of fungal colonization and suggest activation of *BFN1* during root epidermal cell aging/senescence. Next, we inoculated the *pBFN1* reporter lines with *S. indica* and observed a reduction in promoter activity in epidermal cells that were in contact with the fungus compared with mock-treated roots (*Figure 4E and F*). *BFN1* expression and nuclear localization in the root cap or xylem was not affected by *S. indica* colonization (*Figure 4—figure supplement 1*). This indicates that the downregulation of *BFN1* by *S. indica* occurs mainly in epidermal cells of the differentiation zone and is regulated independently of SMB and its activity in the root cap.

To assess the phenotypic effects of *BFN1* downregulation, we analyzed *Arabidopsis bfn1-1* null mutants (*Fendrych et al., 2014*) using the cell death and protein aggregates markers, Evans blue and Proteostat. Staining of *bfn1-1* mutants with Evans blue showed an increase of cell remnants in the epidermal cell layer of the differentiation zone (*Figure 5A and B*), consistent with a proposed delay of dPCD and cell corpse clearance by BFN1 activity. Furthermore, while WT roots were devoid of protein aggregates, *bfn1-1* mutants exhibited aggregates along the primary root axis, starting at

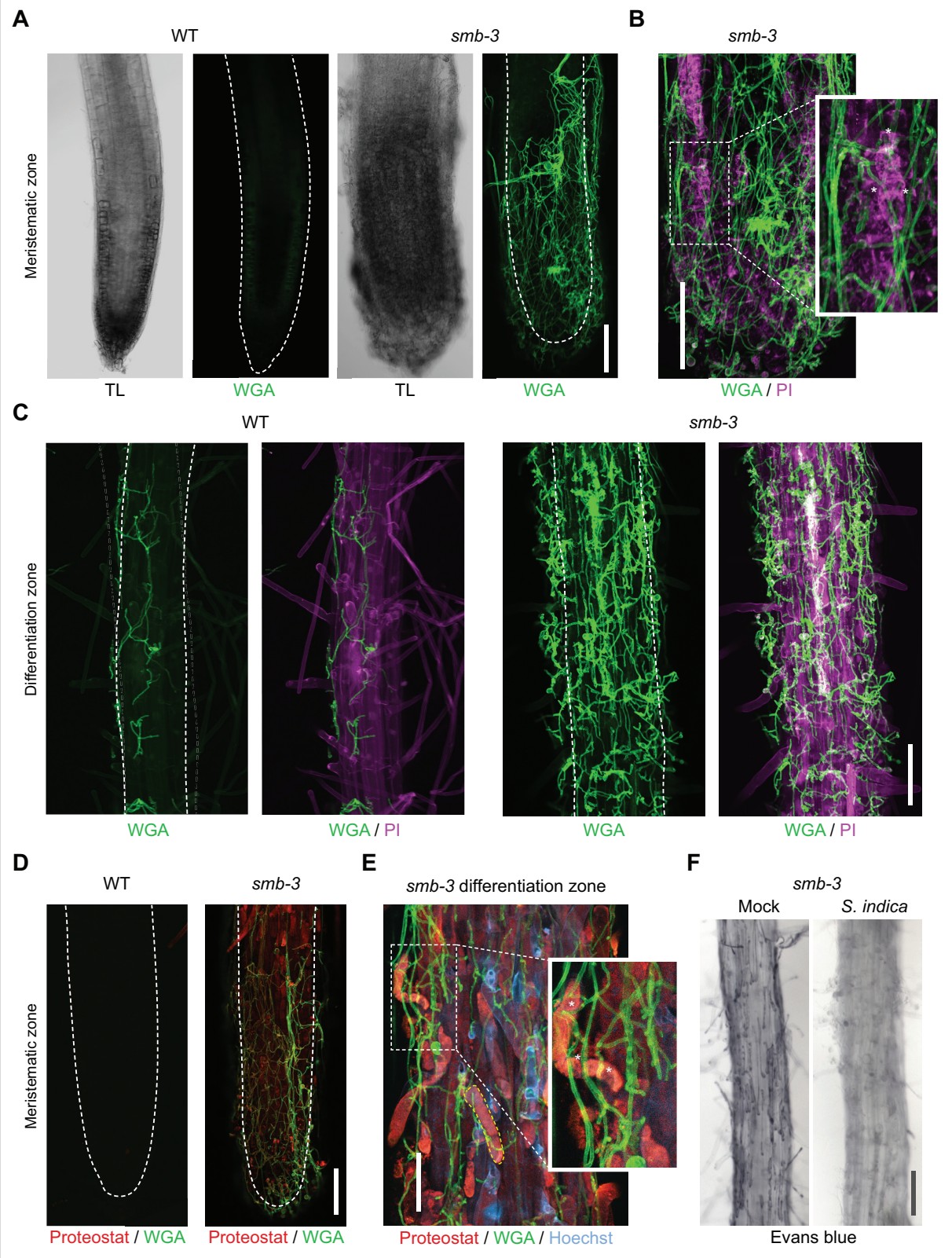

**Figure 3.** Cytological analyses of *S. indica*-colonized *smb-3* mutants and WT roots. For CLSM analyses, 7-day-old seedlings were inoculated with *S. indica* spores and roots were analyzed at 10 dpi. (**A**) Representative images of the meristematic zone of *Arabidopsis* WT and *smb-3* mutants during *S. indica* colonization. WGA-AF 488 stain (green) was used to visualize fungal structures. Transmitted light (TL) images are also shown. Scale indicates 100 µm (**B**) Magnification of a *smb-3* mutant root tip colonized with *S. indica*. Asterisks indicate penetration of hyphae into dead cells stained with

*Figure 3 continued on next page*

*Figure 3 continued*

propidium iodide (PI, Sigma-Aldrich) shown in magenta. Scale indicates 100 µm. (**C**) Representative images of the differentiation zone of WT and *smb-3* mutants colonized with *S. indica* and stained with WGA-AF 488 and PI. Scale indicates 100 µm. (**D**) Representative images of the meristematic zone of WT and *smb-3* mutant root tips inoculated with *S. indica*, stained with WGA-AF 488 and Proteostat (red). Scale indicates 100 µm (**E**) Magnification of the root differentiation zone of *smb-3* mutants showing *S. indica* colonization, stained with WGA-AF 488, Hoechst and Proteostat. Penetration of fungal hyphae into uncleared cell corpses is marked with asterisks. Dotted yellow line indicates lateral root cap (LRC) cell corpse. Scale indicates 50 µm. (**F**) Representative images of the differentiation zone of *S. indica*-colonized WT and *smb-3* roots at 10 dpi, stained with Evans blue. Scale indicates 100 µm.

the transition between elongation and differentiation zone. The meristematic zone remained free of protein aggregates (*Figure 5C and D*). These data suggest that the lack of BFN1 activity in the root cap, xylem, and senescent epidermis creates a general cellular stress in the roots that affects proteostasis in the differentiation zone (*Figure 5—figure supplement 1*). Similar to *bfn1-1*, WT roots colonized by *S. indica* showed increased Evans blue and Proteostat signal in the differentiation zone. Aggregated proteins were detected in colonized and adjacent non-colonized cells along the differentiation zone, suggesting a non-cell autonomous host response to the fungus (*Figure 5—figure supplement 2*).

To investigate the biological relevance of *BFN1* downregulation during *S. indica* root colonization, we quantified extraradical fungal growth using the WGA-AF 488 stain. When comparing staining intensities of *S. indica*-inoculated *bfn1-1* mutants and WT seedlings, we observed a significantly stronger fluorescence signal at the roots of *bfn1-1* mutants, indicating a higher extraradical fungal colonization along the differentiation zone (*Figure 5E and F* and *Figure 5—figure supplement 1*). However, similar to WT roots, *bfn1-1* mutants did not exhibit fungal colonization around the meristematic zone as observed in *smb-3*-colonized roots (*Figure 5E*). Quantification of intraradical colonization by qPCR, after removal of outer fungal mycelium, showed a significant increase of *S. indica* biomass in *bfn1-1* mutants at later stages of interaction (*Figure 5G*). Together, these results emphasize that downregulation of *BFN1* during colonization is beneficial for intra- and extraradical fungal accommodation in the differentiation zone.

Next, we investigated the impact of other beneficial microbes on dPCD by examining transcriptional responses in *Arabidopsis* roots colonized by different organisms. These included *Serendipita vermifera*, an orchid mycorrhizal fungus closely related to *S. indica*, and two bacterial synthetic communities (SynComs) derived from either *Arabidopsis* roots or the rhizosphere of *Hordeum vulgare* (*Mahdi et al., 2022*). In all three interactions, *BFN1* expression was consistently decreased in *Arabidopsis* roots (*Figure 5—figure supplement 3*). Additionally, RNA-Seq analysis of *Arabidopsis* dPCD marker genes during *S. vermifera* colonization confirmed the downregulation of *BFN1* (*Figure 5—figure supplement 3*). Our findings indicate that microbes may benefit from delayed post-mortem corpse clearance after dPCD in host plants and suggest that beneficial microbes may have evolved mechanisms to manipulate dPCD pathways to increase colonization (*Figure 6*).

## Discussion

In this study, we investigated the functional link between dPCD and microbial accommodation in roots. Impaired dPCD in the *Arabidopsis smb-3* and *bfn1-1* mutants increased colonization by the beneficial endophyte *S. indica*. The *smb-3* mutants displayed hypercolonization along the entire primary root, suggesting that the extra sheet of cell corpses surrounding *smb-3* roots provide additional and easily accessible nutrients that fuel fungal colonization. In fact, we observed a clearing effect with progressive colonization stages. Furthermore, hypercolonization of *Arabidopsis*, caused by the loss of dPCD in the root cap, mitigated the beneficial effects of *S. indica* by delaying the induction of growth promotion. Most notably, we observed hypercolonization of the meristematic zone of *smb-3* mutants (*Figure 6*). The root apical meristem embedded in this zone of the root tip is essential for root growth, as all primary root tissue originates from these continuously dividing stem cells (*Hamamoto et al., 2006*). This sensitive tissue is surrounded by the root cap, which protects it from external stresses (*Kumar and Iyer-Pascuzzi, 2020*). The phenotype of *smb-3* mutants shows resemblance to the human skin disease hyperkeratosis. In healthy human skin, a pool of stem cells produces layers of cells that divide, differentiate, die, and are shed. Such developmental programs form a physical and dynamic

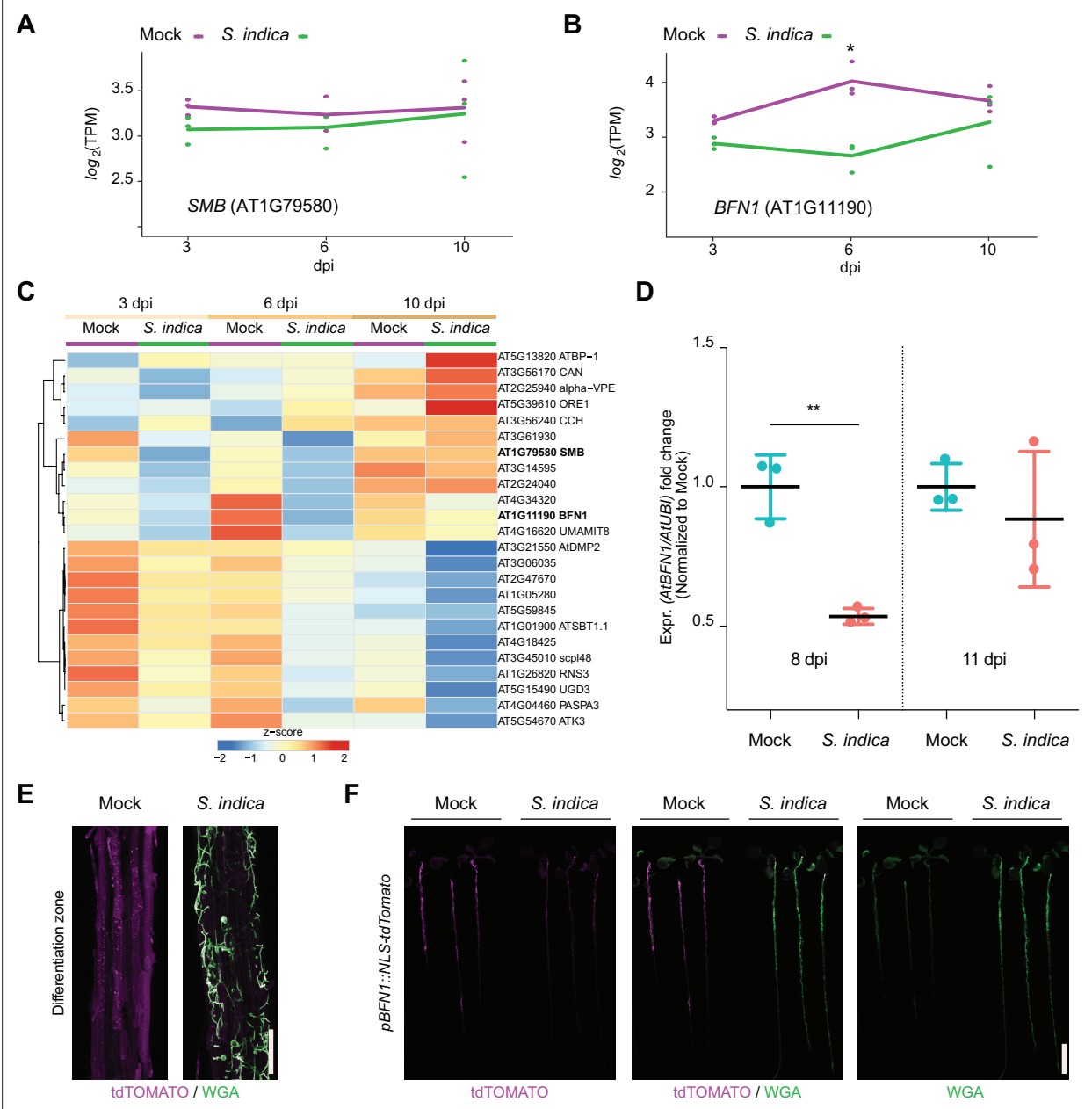

**Figure 4.** *BFN1* is downregulated during interaction with *S. indica*. RNA-Seq expression profiles of (**A**) *SMB* and (**B**) *BFN1* in *Arabidopsis* roots mock-treated or inoculated with *S. indica* at 3, 6, and 10 dpi. The log2-transformed Transcript per Kilobase million (TPM) values are shown and the lines indicate average expression values among the 3 biological replicates. Asterisk indicates significantly different expression (adjusted p-value <0.05) (**C**) The heat map shows the expression values (TPM) of *Arabidopsis* dPCD marker genes with at least an average of 1 TPM across *Arabidopsis* roots mock-treated or inoculated with *S. indica* at 3, 6, and 10 dpi. The TPM expression values are log2 transformed and row-scaled. Genes are clustered using spearman correlation as distance measure. Each treatment displays the average of three biological replicates. The dPCD gene markers were previously defined (***Olvera-Carrillo et al., 2015***). (**D**) *BFN1* expression in WT *Arabidopsis* roots during *S. indica* colonization at 8 and 11 dpi. RNA was isolated from three biological replicates, pooling 30 plants per conditions for qPCR analysis, comparing *BFN1* expression with an *Arabidopsis* ubiquitin marker gene. Statistical significance was determined by one-way ANOVA and Tukey's post hoc test (F [3, 8]=7263; p<0.05). (**E**) Representative CLSM images of the differentiation zone of mock- and *S. indica*- colonized *pBFN1::NLS-tdTOMATO* reporter roots at 7 dpi. The tdTOMATO signal (magenta) represents BFN1 expression and *S. indica* was stained with WGA-AF 488 (green). Scale indicates 100 μm. (**F**) Whole seedling scans of mock- and *S. indica*-treated *pBFN1::NLS-tdTOMATO* plants taken with a LI-COR Odyssey M imager at 7 dpi. Images show *BFN1* expression via tdTOMATO signal in mock conditions and BFN1 expression in presence of *S. indica* (stained with WGA-AF 488). Scale indicates 5 mm.

The online version of this article includes the following figure supplement(s) for figure 4:

**Figure supplement 1.** Expression pattern of *BFN1* in root tissue.

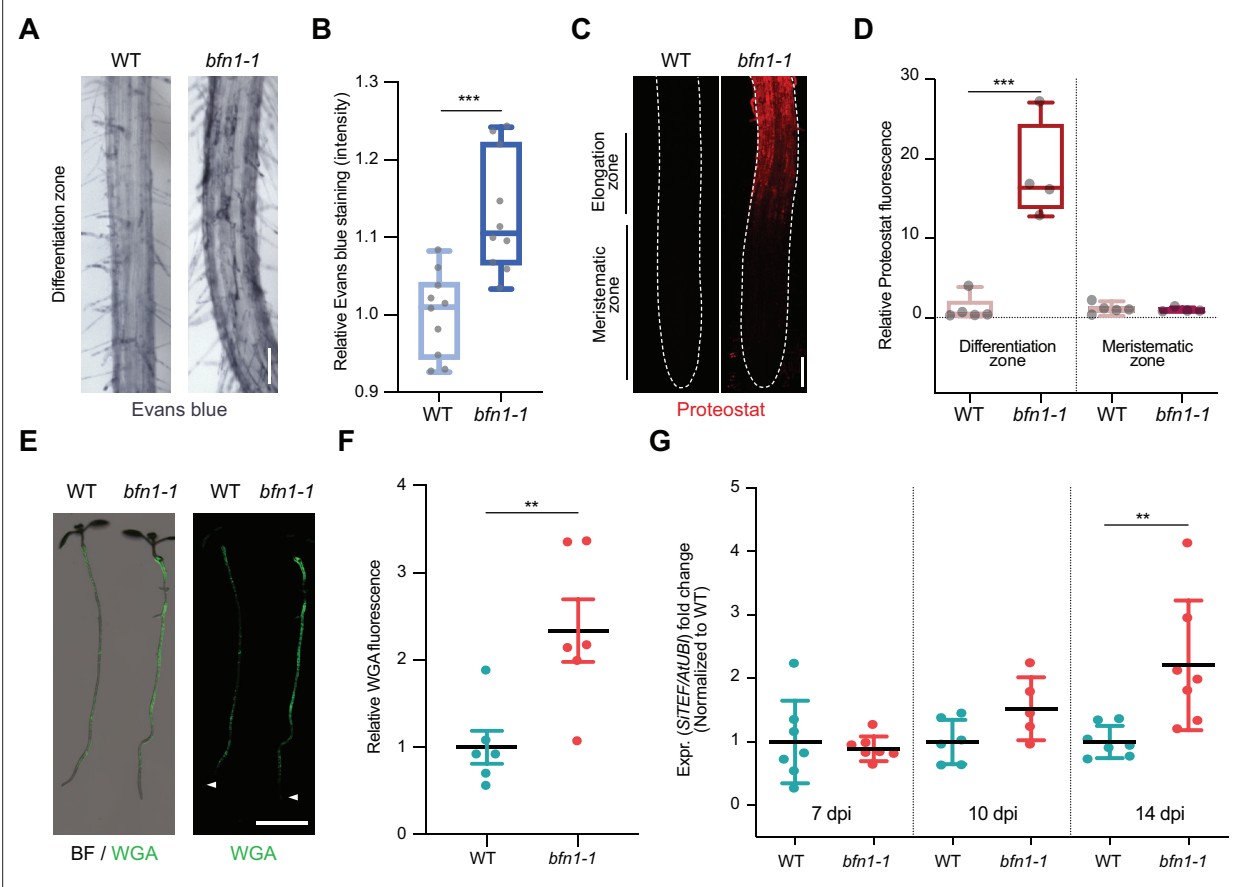

**Figure 5.** *BFN1* downregulation promotes fungal accommodation. (**A**) Microscopy images of the differentiation zone of 14-day-old WT and *bfn1-1* mutant roots, stained with Evans blue. Scale indicates 100 µm. (**B**) Relative quantification of Evans blue staining (refers to image shown in A), comparing 14-day-old WT and *bfn1-1* mutants. Ten plants were used for each genotype. Data were normalized to the WT control. Statistical significance was determined using an unpaired, two-tailed Student's t test (F [9, 9]=2.033; p<0.001). (**C**) Proteostat staining of 10-day-old WT and *bfn1-1* mutant root tips. Scale indicates 100 µm. (**D**) Quantification of Proteostat staining (refers to image shown in C) using 4–5 10-day-old WT and *bfn1-1* mutants. Statistical analysis was performed via one-way ANOVA and Tukey's post hoc test (F [3, 14]=33,55; p<0.001). (**E**) Extraradical colonization of 10-day-old WT and *bfn1-1* mutant plants, seed-inoculated with *S. indica* and stained with WGA-AF 488 (green). Roots were scanned with a LI-COR Odyssey M imager. Arrowheads indicate the position of the uncolonized root tips. Scale indicates 5 mm. (**F**) Relative quantification of extraradical colonization of *bfn1-1* mutant and WT roots, using WGA-AF 488 signal as a proxy for fungal biomass (refers to image shown in E). Data were normalized to the WT control. Statistical comparisons were made by unpaired, two-tailed Student's t test for unpaired samples (F [5, 5]=3.597; p<0.01). (**G**) Intraradical colonization of WT and *bfn1-1* mutants was measured via qPCR. Roots from seven biological replicates were collected and washed to remove outer extraradical mycelium, pooling approximately 30 plants per time point and replicate for each genotype. Each time point was normalized to WT for relative quantification of colonization. Statistical analysis was performed via one-way ANOVA and Tukey's post hoc test (F [5, 33]=5.358; p<0.01).

The online version of this article includes the following figure supplement(s) for figure 5:

**Figure supplement 1.** *bfn1-1* mutant shows increased protein aggregation in roots.

**Figure supplement 2.** Phenotypic analysis of *S. indica* colonization on WT *Arabidopsis* roots.

**Figure supplement 3.** *AtBFN1* is downregulated during colonization with beneficial microbes.

barrier against environmental factors. Microbes attempting to establish themselves are consistently removed by skin exfoliation (*Dettmer, 2021*). Patients with hyperkeratosis show an accumulation of dead cells on the outer skin layer, making them more susceptible to microbial infection (*Cheng et al., 1992*). Likewise, malfunctions within regulated cell death in mammal gut epithelial cells produce death-induced nutrient release (DINNR) that can fuel bacterial growth and infection and could cause a variety of disorders such as inflammatory diseases (*Anderson et al., 2021*). The importance of an intact root cap in plant-microbe interactions is further highlighted by the fact that the physical removal of root caps in maize plants leads to increased colonization of the root tip by the plant growth promoting rhizobacterium (PGPR) *Pseudomonas fluorescens* (*Humphris et al., 2005*) and to changes

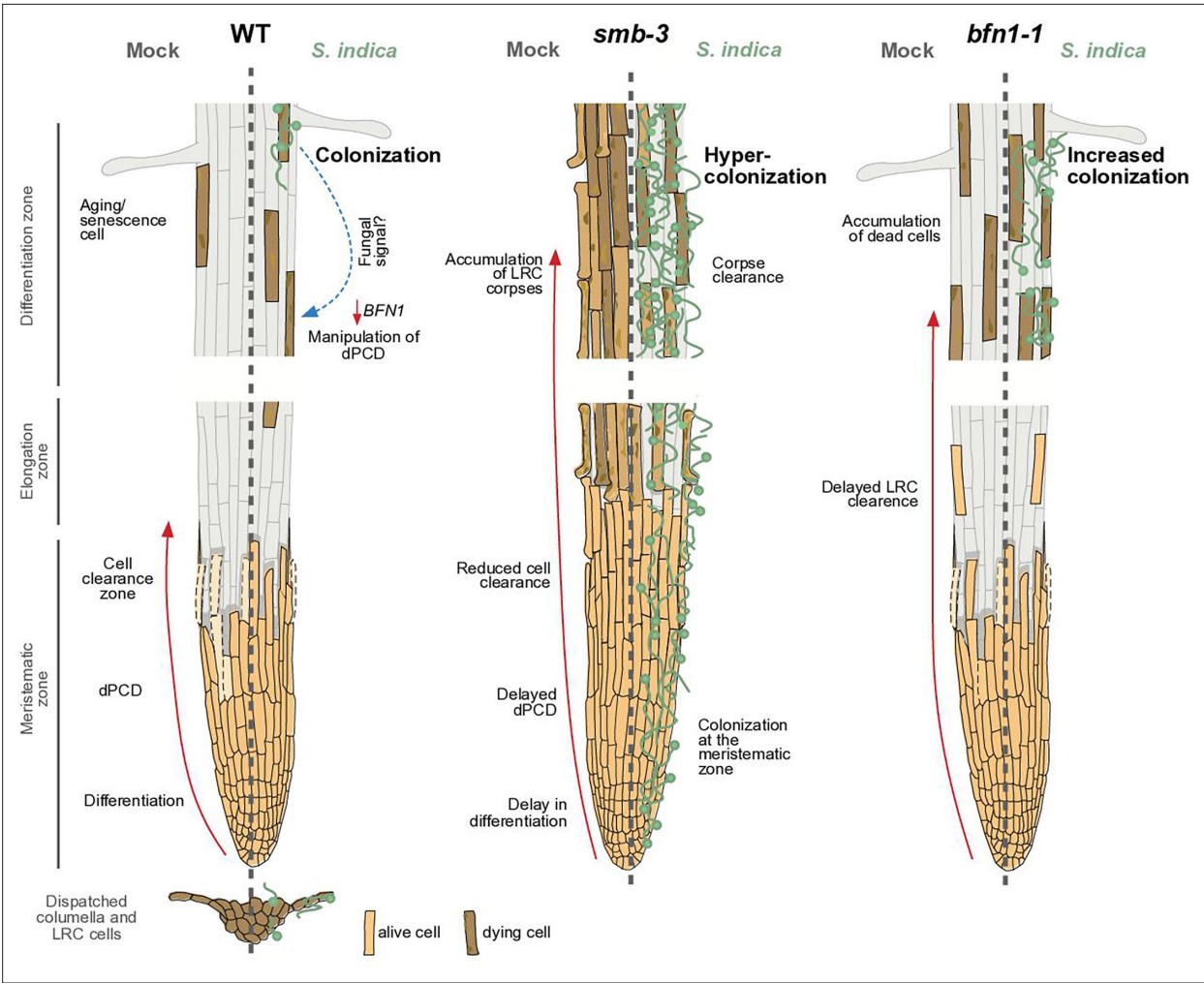

**Figure 6.** dPCD and its proposed effects on plant-microbe interactions. The root cap protects and covers the stem cells of the root apical meristem. Its size in *Arabidopsis* is maintained by a high cellular turnover of root cap cells. While the columella root cap is shed from the root tip, a dPCD machinery marks the final step of LRC differentiation and prevents LRC cells from entering into the elongation zone. Induction of cell death by the transcription factor SMB is followed by irreversible DNA fragmentation and cell corpse clearance, mediated by the nuclease BFN1, a downstream executor of dPCD (*Fendrych et al., 2014*). The absence of dPCD induction in the *smb-3* knockout mutant leads to a delay in LRC differentiation and allows LRC cells to enter the elongation zone, where they die uncontrolled, resulting in an accumulation of LRC cell corpses along the differentiation zone. In a WT background, the fungal endophyte *S. indica* colonizes the differentiation zone of *Arabidopsis* roots and can also be found in shed columella cell packages. The impaired dPCD of the s*mb-3* mutant phenotype results in a hypercolonization of *Arabidopsis* roots, along the differentiation zone as well as the meristematic zone, highlighting that the continuous clearance of root cap cells is necessary for restricting microbial accommodation at the meristematic zone. Loss of the downstream dPCD executor *BFN1* does not affect fungal colonization in the meristematic zone but increases accommodation by *S. indica* in the differentiation zone, where BFN1 appears to be involved in dPCD of senescent epidermal cells and undergoes downregulation during *S. indica* colonization.

in the rhizosphere microbiome composition along the root axis (*Rüger et al., 2023*). Moreover, it has been suggested that border cells released from the root cap may distract root-feeding nematodes from attacking plant roots (*Rodger et al., 2003*). Our results provide strong evidence that root cap size maintenance in the form of constant root cap cell turnover in *Arabidopsis* acts as a dynamic barrier, analogous to epidermal cell turnover in animals. It thus represents a sophisticated physical mechanism to prevent or reduce intracellular microbial colonization near the root meristematic tissue and contributes to the maintenance of a beneficial interaction with root endophytes.

Mutation of *BFN1* displayed significantly increased colonization in the differentiation zone but not in the meristematic zone (*Figure 6*). The differences in colonization patterns to the *smb-3* mutants likely reflect spatial expression and localization patterns of SMB and BFN1 activity throughout *Arabidopsis* roots. While the expression of the transcription factor *SMB* is restricted to the LRC, the

senescence-associated nuclease *BFN1* is expressed in different tissues undergoing dPCD and senescence below and above ground (*Escamez et al., 2020*; *Farage-Barhom et al., 2008*). Here, we show that *BFN1* is additionally expressed in differentiated root epidermal cells that undergo nuclear degradation during root maturation. This suggests an age-dependent dPCD in the outer epidermal layer of *Arabidopsis* roots where expression of *BFN1* possibly pre-dates cortical and epidermal abscission during secondary root growth and contributes to clearance of cell corpses during periderm emergence (*Wunderling et al., 2018*). This process resembles root cortical senescence and cell death in grass species such as wheat, barley, and corn, which typically start in the epidermis and spread toward the endodermis (*Drew et al., 2000*).

Additionally, we observed distinct differences in the presence and distribution of protein aggregates in *smb-3* and *bfn1-1* mutants. In *smb-3* mutants, protein aggregates are restricted to LRC cells adhering to the primary root. This localized distribution may be attributed to the absence of SMB-induced proteases in the LRC cells of these mutants. The absence of these proteases likely impairs protein turnover and degradation, leading to accumulation and aggregation specifically in these cells. In contrast, *bfn1-1* mutation results in a more extensive and diffuse pattern of protein aggregation in the epidermal cell layer of the differentiation zone, regardless of the occurrence of cell death. This widespread distribution suggests that BFN1 plays a broader role in maintaining cellular homeostasis and protein quality control throughout the root epidermis and differentiation zone. These distinct patterns of protein aggregation and cell death in *smb-3* and *bfn1-1* mutants underscore the importance of these genes in maintaining cellular integrity and protein homeostasis. The contrasting phenotypes highlight the distinct roles of these genes in cellular processes.

While we show that dPCD protects the meristem from microbial colonization, we propose that some adapted microbes manipulate host dPCD processes by affecting the transcriptional expression of *BFN1* to facilitate accommodation in the root. Whether active interference by fungal effector proteins, fungal-derived signaling molecules or a systemic response of *Arabidopsis* roots underlies *BFN1* downregulation by *S. indica* remains to be investigated. It was recently demonstrated that small active metabolites produced either by Toll/interleukin-1 receptor (TIR)-containing leucine-rich repeat (NLR) receptors (*Yu et al., 2022*) or by fungal-derived enzymes (*Dunken et al., 2022*) through hydrolysis of RNA/DNA can mediate host cell death. This raises an exciting possibility that RNAse and DNAse activities of BFN1 may be involved in producing small, active nucleotide-derived metabolites that affect cell death, cell corpse clearance, proteostasis and fungal accommodation in the differentiation zone. To explore this hypothesis further, metabolomic and proteomic approaches should be employed in future studies. Notably, we have shown that colonization by other beneficial microbes such as the closely related fungus *S. vermifera* and bacterial members of the *Arabidopsis* and *H. vulgare* microbiota (*Mahdi et al., 2022*) also led to the downregulation of *BFN1* in *Arabidopsis*. These findings emphasize the presence of a conserved pathway influenced by diverse beneficial microbes to downregulate *BFN1* expression in epidermal tissue to facilitate symbioses.

Transcriptomic analysis of both established and predicted key dPCD marker genes revealed diverse patterns of upregulation and downregulation during *S. indica* colonization. These findings provide a valuable foundation for future studies investigating the dynamics of dPCD processes during beneficial symbiotic interactions and the potential manipulation of these processes by symbiotic partners.

In conclusion, our data show that tight regulation of host dPCD in epidermal- and root cap-tissue plays an important role in restricting fungal colonization. From a microbial perspective, dPCD pathways represent a potential nexus for enhancing symbiotic interactions and potentially improving nutrient availability (*Figure 6*). These results shed light on the complex relationship between PCD and microbial accommodation in plant roots, offering valuable insights into the development of plants that establish more efficient partnerships with beneficial microbes.

# Materials and methods

**Key resources table**

| Reagent type (species) or resource | Designation | Source or reference | Identifiers | Additional information |
|---|---|---|---|---|
| Strain, strain background (*Serendipita indica*) | DSM11827 | German 362 Collection of Microorganisms and Cell Cultures, Braunschweig, Germany | DSM11827 | |
| Strain, strain background (*Arabidopsis thaliana*) | WT Col-0 | N-60000 | Col-0 | |
| Genetic reagent (*A. thaliana*) | *bfn1-1* | **Fendrych et al., 2014** | GK-197G12 | |
| Genetic reagent (*A. thaliana*) | *smb-3* | **Fendrych et al., 2014** | SALK_143526 C | |
| Genetic reagent (*A. thaliana*) | *pBFN1::NLS-tdTOMATO* | **Huysmans et al., 2018** | | |
| Commercial assay, kit | Proteostat Aggresome detection kit | Enzo Life Sciences | CAT# ENZ-51035–0025 | 0.5 µl/ml Proteostat and 0.5 µl/ml Hoechst 33342 |
| Antibody | anti-polyQ (Mouse monoclonal) | Merck | MAB1574 | 1:1000 |
| Antibody | anti-Actin (Rabbit polyclonal) | Agrisera | AS132640 | 1:5000 |
| Software, algorithm | Prism8 | https://www.graphpad.com/ | RRID:SCR_002798 | |
| Software, algorithm | ImageJ | https://imagej.net/ | RRID:SCR_003070 | |
| Software, algorithm | EmpiriaStudio Software | LI-COR Biosciences | RRID:SCR_022512 | v. 3.2.0.186 |
| Software, algorithm | LASX | https://www.leica-microsystems.com/ | RRID:SCR_013673 | |
| Sequence-based reagent | AtUBI_F | This paper | PCR primers | CCAAGCCGAAGAAGATCAAG |
| Sequence-based reagent | AtUBI_R | This paper | PCR primers | ACTCCTTCCTCAAACGCTGA |
| Sequence-based reagent | FW_BFN1-A_qPCR | This paper | PCR primers | GGCGTCAAGTCTGGTGAAAC |
| Sequence-based reagent | RV_BFN1-A_qPCR | This paper | PCR primers | ACCCGGTTTAGTATCATGGCT |
| Sequence-based reagent | TEF_S. indica qPCR_F | This paper | PCR primers | GCAAGTTCTCCGAGCTCATC |
| Sequence-based reagent | TEF_S. indica qPCR_R | This paper | PCR primers | CCAAGTGGTGGGTACTCGTT |
| Other | Odyssey M Imaging System | LI-COR Biosciences | | Equipment |
| Other | FV1000 confocal laser scanning microscope | Olympus | | Equipment |
| Other | LSM Meta 710 | Carl Zeiss Technology | | Equipment |
| Other | Leica M165 FC Microscope | Leica Microsystems | | Equipment |
| Other | CFX connect real time system | BioRad | | Equipment |

*Continued on next page*

*Continued*

| Reagent type (species) or resource | Designation | Source or reference | Identifiers | Additional information |
|---|---|---|---|---|
| Other | Alexa Fluor 488 conjugated with Wheat Germ Agglutinin (WGA-AF) stain | Invitrogen | CAT# 11261 | 5 μl/mL from 1 mg/ml stock |
| Other | Propidium iodide (PI) stain | Sigma-Aldrich | | 10 μg/ml |
| Other | Evans blue stain | Sigma-Aldrich | CAS# 314-13-6 | 2 mM Evans blue staining concentration |

## Fungal strains and growth conditions

Fungal experiments were performed with *Serendipita indica* strain DSM11827 (German Collection of Microorganisms and Cell Cultures, Braunschweig, Germany). *S. indica* was grown on complete medium (CM) containing 2% (w/v) glucose and 1.5% (w/v) agar (*Hilbert et al., 2012*). Fungal material was grown at 28 °C in the dark for 4 weeks before spore preparation. For additional experiments, *S. vermifera* (MAFF305830) was used and grown on MYP medium 7 g/l malt extract (Sigma-Aldrich), 1 g/l peptone (Sigma-Aldrich), 0.5 g/l yeast extract (Carl Roth) containing 1.5% agar at 28 °C in darkness for 3 weeks before mycelium preparation for root inoculation.

## Plant material and growth conditions

Seeds of *Arabidopsis thaliana* wild-type (WT) ecotype Columbia 0 (Col-0) and T-DNA insertion mutants (*bfn1-1* [GK-197G12] and *smb-3* [SALK_143526 C]) in Col-0 background were used for experiments. Seeds were surface sterilized in 70% ethanol for 15 min and 100% ethanol for 12 min, stratified at 4 °C in the dark for 3 days and germinated and grown on ½ MS medium (Murashige-Skoog Medium, with vitamins, pH 5.7, Duchefa Biochemie) containing 1% (w/v) sucrose and 0.4% (w/v) Gelrite (Duchefa Biochemie) under short-day conditions (8 hr light, 16 hr dark) with 130 μmol $m^{-2 s-1}$ light and 22 °C/18 °C.

## Fungal inoculation

One-week-old seedlings were transferred to 1/10 PNM (Plant minimal Nutrition Medium, 0.5 mM $KNO_3$, 0.367 mM $KH_2PO_4$, 0.144 mM $K_2HPO_4$, 2 mM $MgSO_4$ x $H_2O$, 0.2 mM $Ca(NO_3)_2$, 0.25% (v/v) Fe-EDTA (0.56% w/v $FeSO_4$ x$7H_2$ O and 0.8% w/v $Na_2EDTA$ x $2H_2O$), 0.428 mM NaCl; pH-adjusted to 5.7 and buffered with 10 mM MES. For solid media, 0.4% (w/v) Gelrite (Duchefa Biochemie) was added) plates without sucrose, using 15–20 seedlings per plate. Under sterile conditions, spores of *S. indica* were scraped from agar CM plates using 0.002% Tween water (Roth), washed two times with dd$H_2O$ and pipetted in a volume of 2 ml on plant roots and surrounding area in a concentration of $5 \times 10^5$ spores per plate. dd$H_2O$ was used for inoculation of mock plants. For *S. vermifera* inoculation, mycelium was scrapped from plates in dd$H_2O$, washed and added to *Arabidopsis* roots in a volume of 1 ml of a stock solution of 1 g / 50 ml.

In case of experiments using seeds inoculation with *S. indica*, *Arabidopsis* seeds were surface sterilized, incubated with fungal spore solution at $5 \times 10^5$ concentration for 1 hr and plated on ½ MS plates (without sucrose).

## Evans blue staining

For the visualization of cell death in *Arabidopsis* roots, a modified protocol by *Vijayaraghavareddy et al., 2017* was used. Roots were washed three times in dd$H_2O$ to remove loose external fungal mycelium and then stained for 15 min in 2 mM Evans blue (Sigma-Aldrich) dissolved in 0.1 M $CaCl_2$ pH 5.6. Subsequently, roots were washed extensively with dd$H_2O$ for 1 hr and a Leica M165 FC microscope was used for imaging. Four pictures were taken along the main root axis of each plant and averaged together, for an overview of cell death in the differentiation zone of one root. To quantify Evans blue staining intensity, ImageJ was used to invert the pictures, draw out individual roots and measure and compare mean grey values.

## Extraradical colonization assays

To quantify extraradical colonization of *S. indica* on *Arabidopsis*, seed-inoculated plants were grown for 10 days. Inoculated and mock-treated seedlings were stained directly on agar plate by pipetting 2 ml of 1 X Phosphate-buffered saline (PBS, 137 mM NaCl, 2.7 mM KCl, 10 mM $Na_2HPO_4$, 1.8 mM $KH_2PO_4$) solution containing Alexa Fluor 488 conjugated with Wheat Germ Agglutinin (5 μl/mL from 1 mg/ml stock; WGA-AF 488, Invitrogen). After 2 min of incubation, the roots were washed twice on the plate with 1 X PBS solution. The stained seedlings were transferred to a fresh ½ MS square plate (Greiner Bio-One). In order to perform a correct and focused scan of the agar plate with the roots, it was checked that the solid MS medium was flat and even and had no unevenness. To scan the agar plate, we used an Odyssey M Imaging System (LI-COR Biosciences) and the LI-COR Acquisition Software 1.1 (LI-COR Biosciences). In the software, we selected custom assay and then membrane. We selected the area in the agar plate to be scanned and selected the channels 488 (for WGA-Alexa Flour 488) and RGB trans (for bright field). We defined the focus offset between 3.5 mm and 4.0 mm (depending on the thickness of the MS medium). For resolution we selected 10 μm when scanning two plants or 100 μm when scanning several plants. Quantification of WGA-AF 488 fluorescence was performed using EmpiriaStudio Software (LI-COR Biosciences).

## RNA extraction (intraradical colonization assay)

To measure intraradical colonization via RNA extraction and PCR, plants were harvested at three time points around 7, 10, and 14 dpi. The roots were extensively washed with $ddH_2O$ and tissue paper was used to carefully wipe off mycelium on the surface of the roots. After cleaning, the roots were shock frozen in liquid nitrogen and RNA was extracted with TRIzol (Invitrogen, Thermo Fisher Scientific, Schwerte, Germany). After a DNase I (Thermo Fisher Scientific) treatment according to the manufacturer's instructions to remove DNA, RNA was used to generate cDNA through the utilization of the Fermentas First Strand cDNA Synthesis Kit (Thermo Fisher Scientific).

## Quantitative RT-PCR analysis

The quantitative real-time PCR (qRT-PCR) was performed using a CFX connect real time system (BioRad) with the following program: 95 °C 3 min, 95 °C 15 s, 59 °C 20 s, 72 °C 30 s, 40 cycles and melting curve analysis. Relative expression was calculated using the $2^{-\Delta\Delta CT}$ method (*Livak and Schmittgen, 2001*). qRT-PCR primers can be found in the Key Resources Table.

## Filter trap analysis

Filter trap assays were performed as previously described (*Llamas et al., 2023*; *Llamas et al., 2021*). Protein extracts were obtained using native lysis buffer (300 mM NaCl, 100 mM HEPES pH 7.4, 2 mM EDTA, 2% Triton X-100) supplemented with 1 x plant protease inhibitor (Merck). Cell debris was removed by several centrifugation steps at 8000 x *g* for 10 min at 4 °C. The supernatant was separated, and protein concentration determined using the Pierce BCA Protein Assay Kit (Thermo Fisher). A cellulose acetate membrane filter (GE Healthcare Life Sciences) and 3 filter papers (BioRad, 1620161) were immersed in 1 x PBS and placed in a slot blot apparatus (Bio-Rad) connected to a vacuum system. The membrane was equilibrated by 3 washes with equilibration buffer (native buffer containing 0.5% SDS). 300, 200, and 100 μg of the protein extract were mixed with SDS at a final concentration of 0.5% and filtered through the membrane. The membrane was then washed with 0.2% SDS and blocked with 3% BSA in Tris-Buffered Saline 0.1% Tween (TBST, 50 mM Tris-CL, 150 mM NaCl, pH 7.5) for 30 minutes, followed by three washes with TBST. Incubation was performed with anti-polyQ [1:1000] (Merck, MAB1574). The membrane was washed 3 times for 5 min and incubated with secondary antibodies in TBST 3% BSA for 30 min. The membrane was developed using the Odyssey M Imaging System (LI-COR Biosciences). Extracts were also analyzed by SDS-PAGE and western blotting against anti-Actin [1:5000] (Agrisera, AS132640) to determine loading controls.

## Confocal laser scanning microscopy (CLSM) and proteostat staining quantification

CLSM images were acquired using either the FV1000 confocal laser scanning microscope (Olympus) or a Meta 710 confocal microscope with laser ablation 266 nm (Zeiss). All images were acquired using the same parameters between experiments. Excitation of WGA-AF 488 was done with an

argon laser at 488 nm and the emitted light was detected with a hybrid detector at 500–550 nm. Proteostat was excited at 561 nm and the signal was detected between 590 and 700 nm. Hoechst was excited with a diode laser at 405 nm and the emitted light was detected with a hybrid detector at 415–460 nm.

## Proteostat staining

For the detection of aggregated proteins, we used the Proteostat Aggresome detection kit (Enzo Life Sciences). Seedlings were stained according to the manufacturer's instructions. Seedlings were incubated with permeabilizing solution (0.5% Triton X-100, 3 mM EDTA, pH 8.0) for 30 minutes at 4 °C with gentle shaking. The seedlings were washed twice with 1 X PBS. Then the plants were incubated in the dark with 1 x PBS supplemented with 0.5 µl/ml Proteostat and 0.5 µl/ml Hoechst 33342 (nuclear stain) for 30 min at room temperature. Finally, the seedlings were washed twice with 1 x PBS and mounted on a slide for CLSM analysis or in mounted in fresh MS phytoagar plates for LI-COR analysis. Quantification of Proteostat fluorescence was performed using Fiji software or EmpiriaStudio Software (LI-COR Biosciences).

## Transcriptomic analysis

*Arabidopsis* Col-0 WT roots were inoculated with *S. indica*. *Arabidopsis* roots were harvested from mock-treated plants and inoculated plants at four different time points post inoculation: 1, 3, 6, and 10 dpi. Three biological replicates were used for each condition. The RNA-seq libraries were generated and sequenced at US Department of Energy Joint Genome Institute (JGI) under a project proposal (Proposal ID: 505829) (*Eichfeld et al., 2024*; *Zuccaro and Langen, 2020*). For each sample, stranded RNA-Seq libraries were generated and quantified by qRT-PCR. RNA-Seq libraries were sequenced with Illumina sequencer. Raw reads were filtered and trimmed using the JGI QC pipeline. Filtered reads from each library were aligned to the *Arabidopsis* genome (TAIR10) using HISAT2 (*Kim et al., 2015*) and the reads mapped to each gene were counted using featureCounts (*Liao et al., 2014*). Samples harvested at 1 dpi were omitted from the analysis because we decided to focus on the time points at which the interaction between *Arabidopsis* and *S. indica* is well established. Differential gene expression analysis was performed using the R package DESeq2 (*Love et al., 2014*). Genes with an FDR adjusted p-value <0.05 were considered as differentially expressed genes (DEGs). The adjusted p-value refers to the transformation of the p-value obtained with the Wald test after considering multiple testing. To visualize gene expression, genes expression levels were normalized as Transcript Per kilobase million (TPM).

## Statistical analyses

All statistical analyses, except for the transcriptomic analysis, were performed using Prism8. Individual figures state the applied statistical methods, as well as p and F values. p-*values* and corresponding asterisks are defined as following, $p<0.05$ *, $p<0.01$**, $p<0.001$***.

## Acknowledgements

We thank the imaging facilities of CECAD (A Schauss and C Jüngst) and CEPLAS (PS Tan) for their assistance with CLSM. We thank Lisa Mahdi for conducting the experiments and providing the samples used for the RNA-seq analysis. We further would like to thank Yu Zhang, Sravanthi Tejomurthula, Daniel Peterson, Vivian Ng & Igor Grigoriev and their work performed in the work proposal (10.46936/10.25585/60001292) conducted by the U.S. Department of Energy Joint Genome Institute (https://ror.org/04xm1d337), a DOE Office of Science User Facility, is supported by the Office of Science of the U.S. Department of Energy operated under Contract No. DE-AC02-05CH11231. AZ, NC and EL acknowledge support by the German Research Foundation (DFG) - Excellence Strategy of the Federal Republic of Germany - EXC-2048/1 - project ID 390686111. AZ and NC acknowledge support by the SFB-1403–414786233 and DV by the German Excellence Strategy-CECAD, EXC 2030–390661388. MKN gratefully acknowledges funding from the European Research Council (ERC) (StG PROCELLDEATH 639234 and CoG EXECUT.ER 864952) as well as the Research Foundation – Flanders (FWO) projects no. G002120N and G002620N.

## Additional information

### Funding

| Funder | Grant reference number | Author |
| --- | --- | --- |
| Deutsche Forschungsgemeinschaft | Excellence Strategy of the Federal Republic of Germany - EXC-2048/1 - project ID 390686111 | Nyasha Charura Ernesto Llamas Concetta De Quattro Alga Zuccaro |
| Deutsche Forschungsgemeinschaft | SFB-1403-414786233 | Nyasha Charura Ernesto Llamas Alga Zuccaro |
| European Research Council | StG PROCELLDEATH 639234 | Moritz K Nowack |
| European Research Council | CoG EXECUT.ER 864952 | Moritz K Nowack |
| Fonds Wetenschappelijk Onderzoek | G002120N | Moritz K Nowack |
| Fonds Wetenschappelijk Onderzoek | G002620N | Moritz K Nowack |
| Deutsche Forschungsgemeinschaft | German Excellence Strategy-CECAD - EXC 2030-390661388 | David Vilchez |

The funders had no role in study design, data collection and interpretation, or the decision to submit the work for publication.

### Author contributions

Nyasha Charura, Ernesto Llamas, Concetta De Quattro, Conceptualization, Data curation, Formal analysis, Validation, Investigation, Visualization, Methodology, Writing – original draft, Writing – review and editing; David Vilchez, Resources, Investigation, Writing – review and editing; Moritz K Nowack, Conceptualization, Resources, Funding acquisition, Investigation, Writing – review and editing; Alga Zuccaro, Conceptualization, Resources, Supervision, Funding acquisition, Investigation, Writing – original draft, Project administration, Writing – review and editing

### Author ORCIDs

Nyasha Charura ⓘ https://orcid.org/0000-0003-4557-8618
Ernesto Llamas ⓘ https://orcid.org/0000-0001-9262-2402
David Vilchez ⓘ https://orcid.org/0000-0002-0801-0743
Moritz K Nowack ⓘ https://orcid.org/0000-0001-8918-7577
Alga Zuccaro ⓘ https://orcid.org/0000-0002-8026-0114

Reviewer #1 (Public Review): https://doi.org/10.7554/eLife.96266.3.sa1
Reviewer #2 (Public Review): https://doi.org/10.7554/eLife.96266.3.sa2
Author response https://doi.org/10.7554/eLife.96266.3.sa3

# Additional files

### Supplementary files

• MDAR checklist

### Data availability

All data generated or analyzed during this study are included in the manuscript and supporting files.

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
