## [Editor Report · eLife assessment]

This study investigated the involvement of programmed cell death (PCD) in *Arabidopsis thaliana* root cap cells and its effect on microbial colonization. The authors have reported the importance of timely corpse clearance in the root cap and a root cap-specific transcription factor in controlling microbial colonization by beneficial fungi. By demonstrating the connection between transcriptional control of PCD and microbial colonization, this study provides **fundamental** insights into how relationships are established and regulated at the root-microbiome interface. The strength of the evidence presented is **convincing**, providing a foundation for further research concerning the spatial and temporal dynamics of microbiome recruitment along the root axis.

---

## [Referee Report · Reviewer #1 (Public Review)]

Summary:

The study investigated how root cap cell corpse removal affects the ability of microbes to colonize *Arabidopsis thaliana* plants. The findings demonstrate how programmed cell death and its control in root cap cells affect the establishment of symbiotic relationships between plants and fungi. Key details on molecular mechanisms and transcription factors involved are also given. The study suggests reevaluating microbiome assembly from the root tip, thus challenging traditional ideas about this process. While the work presents a key foundation, more research along the root axis is recommended to gain a better understanding of the spatial and temporal aspects of microbiome recruitment.

Comments on revised version:

The authors have positively addressed all the critical points I raised in the previous review.

---

## [Referee Report · Reviewer #2 (Public Review)]

Summary:

The authors identify the root cap as an important key region for establishing microbial symbioses with roots. By highlighting for the first time the crucial importance of tight regulation of a specific form of programmed cell death of root cap cells and the clearance of their cell corpses, they start unraveling the molecular mechanisms and its regulation at the root cap (e.g. by identifying an important transcription factor) for the establishment of symbioses with fungi (and potentially also bacterial microbiomes).

Strengths:

It is often believed that the recruitment of plant microbiomes occurs from bulk soil to rhizosphere to endosphere. These authors demonstrate that we have to re-think microbiome assembly as a process starting and regulated at the root tip and proceeding along the root axis.

Comments on revised version:

The authors have addressed all critical points in their revision.

---

## [Author Response]

The following is the authors’ response to the original reviews.

**Public Reviews:**

**Reviewer #1 (Public Review):**
Summary:The study investigated how root cap cell corpse removal affects the ability of microbes to colonize *Arabidopsis thaliana* plants. The findings demonstrate how programmed cell death and its control in root cap cells affect the establishment of symbiotic relationships between plants and fungi. Key details on molecular mechanisms and transcription factors involved are also given. The study suggests reevaluating microbiome assembly from the root tip, thus challenging traditional ideas about this process. While the work presents a key foundation, more research along the root axis is recommended to gain a better understanding of the spatial and temporal aspects of microbiome recruitment.

We thank Reviewer #1 for their positive evaluation and critical feedback.

**Reviewer #2 (Public Review):**
Summary:The authors identify the root cap as an important key region for establishing microbial symbioses with roots. By highlighting for the first time the crucial importance of tight regulation of a specific form of programmed cell death of root cap cells and the clearance of their cell corpses, they start unraveling the molecular mechanisms and its regulation at the root cap (e.g. by identifying an important transcription factor) for the establishment of symbioses with fungi (and potentially also bacterial microbiomes).Strengths:It is often believed that the recruitment of plant microbiomes occurs from bulk soil to rhizosphere to endosphere. These authors demonstrate that we have to re-think microbiome assembly as a process starting and regulated at the root tip and proceeding along the root axis.Weaknesses:The study is a first crucial starting point to investigate the spatial recruitment of beneficial microorganisms along the root axis of plants. It identifies e.g. an important transcription factor for programmed cell death, but more detailed investigations along the root axis are now needed to better understand - spatially and temporally - the orchestration of microbiome recruitment.

We appreciate Reviewers #2 insightful comments and agree that further investigations are needed to gain a deeper understanding of the intricate interplay between the spatial and temporal recruitment of the microbiome and developmental cell death in future studies.

**Recommendations for the authors:**

**Reviewer #1 (Recommendations For The Authors):**
- Given that the smb-3 altered PCD phenotype has already been reported in several publications, the aim of using Evans blue staining to highlight LRC cell corpses along the root surface of smb-3 is not clear. Maybe S1 would be more informative as main figure.

As an indicator of membrane integrity loss and cell death, Evans blue staining was used to characterize all dPCD mutants described in this study and their interactions with *S. indica*. To avoid redundancies with other publications, we restructured Figure 1, incorporating panel S1A to provide an introductory overview of the *smb-3* phenotype. The former Figure 1B is now located in Figure S1.

- It is not clear how the analysis of protein aggregates fits into the rationale, why analyze these formations? What role should they have in the process of PCD or interaction with microbes?

The manuscript has been modified the following way to clarify the analysis of protein aggregates in the dPCD mutants: “The transcription factor SMB promotes the expression of various dPCD executor genes, including proteases that break down and clear cellular debris and protein aggregates following cell death induction. In the LRCs of *smb-3* mutants, the absence of induction of these proteases potentially explains the accumulation of protein aggregates in uncleared dead LRC cells.”.

- Is the accumulation of misfolded and aggregated proteins also present during physiological PCD of LRC cells in the WT?

The biochemical mechanisms underlying PCD can vary depending on the affected cell types and tissues. Within the root tip of Arabidopsis, two different modes of PCD have been described, differentiating between columella root cap cells and LRC cells. For clarification the manuscript has been adjusted the following way:” Under physiological conditions in WT roots, we previously observed protein aggregate accumulation in sloughed columella cell packages, but not during dPCD of distal LRC clearance (Llamas et al., 2021). This aligns with the findings that dPCD of the columella is affected by the loss of autophagy, while dPCD of the LRC is not (Feng et al., 2022).”.

- I suggest being more careful when using the term "root cap" instead of "LRC" to reduce ambiguity (i.e. lines 56; 137), maybe you need to double-check the text.

We agree with the reviewer that a clear distinction between “root cap” and “LRC” is very important. We have adjusted the manuscript to avoid any misunderstandings.

- A technical question regarding qPCR sample preparation: doesn't washing the smb-3 roots cause a loss of LRC stretched cells and would it therefore lead to an alteration of the results?

The mechanical washing of roots is essential to ensure a clear distinction between intraradical fungal growth and accommodation around roots. While we cannot exclude the possibility that mechanical washing removes LRC cells, intraradical quantification of fungal biomass aims to measure *S. indica* growth in the epidermal and cortical cell layers, underneath the uncleared LRC cells. Thus, we complemented this assay with extraradical colonization assays to quantify external fungal biomass with intact LRC cells.

- It is not clear if S. indica promotes PCD in wt and/or in smb-3, could you comment on it?

It remains an open question whether and to what extent *S. indica* promotes PCD, although there are strong indications that this fungus activates different cell death pathways at various developmental stages, including dAdo mediated cell death. We posit that certain microbes have evolved to regulate and manipulate different dPCD processes to enhance colonization, implicating a complex crosstalk between various PCD pathways. We have adjusted the manuscript to underscore this perspective the following way:” Transcriptomic analysis of both established and predicted key dPCD marker genes revealed diverse patterns of upregulation and downregulation during S. indica colonization. These findings provide a valuable foundation for future studies investigating the dynamics of dPCD processes during beneficial symbiotic interactions and the potential manipulation of these processes by symbiotic partners.”.

- How analysis of BFN1 expression in whole root confirms its downregulation at the onset of cell death in S. indica-colonized plants. Moreover, is the transcriptional regulation of BFN1 important for PCD, or is the BFN1 protein level correlated with the establishment of cell death?

*BFN1* gene expression in Arabidopsis shows a transient decrease around 6–8 days after *S. indica* inoculation, coinciding with the proposed onset of *S. indica*-induced cell death. While we can only speculate on a potential correlation between *BFN1* downregulation and the onset of *S. indica*-induced cell death, we have described other pathways through which *S. indica* induces cell death. For example, it produces small metabolites such as dAdo through the synergistic activity of two secreted fungal effector proteins (Dunken et al., 2023). This suggests that *S. indica* recruits different pathways to induce cell death, which may vary depending on the host plant and interact with each other as shown for many other immunity related cell death pathways which share some components.

Regarding the second part of the question, *BFN1* expression correlates positively with cells primed for dPCD (Olvera-Carrillo et al., 2015). BFN1 protein accumulates in the ER lumen and is released into the cytoplasm upon cell death induction to exert its DNase functions (Fendrych et al., 2014). If accumulation of BFN1 is cause or consequence of cell death remains to be validated.

- Line 190: there is a typo "in the nucleus", this is superfluous given that the reporter is nuclear.

The manuscript has been adjusted accordingly; see line L208. However, we consider the distinction important as we aim to emphasize the difference between the nuclear localization of the fluorescent signal in "healthy" cells and the dispersed fluorescent signal spreading in the cytoplasm of cells priming or undergoing dPCD.

- Line 255: there is a typo, stem cells can not differentiate.

The manuscript has been adjusted.

- During root hair development some epidermal cells undergo PCD to allow the emergence of root hairs. Furthermore, during plant defense against pathogens, epidermal cells undergo cell death to prevent further colonization. Have these cell death events been reported to occur under physiological conditions during development?

Plant defence responses in roots and the hypersensitive response (HR) still remain largely unexplored. The HR is a defence mechanism that consists of a localized and rapid cell death at the site of pathogen invasion. It is triggered by pathogenic effector proteins, usually recognized by intracellular immune receptors (NLRs), and accompanied by other features such as ROS signalling, Ca2+ bursts and cell wall modifications (Balint-Kurti, 2019). Notably, HR has been widely described in leaves, but no strong evidence has been shown for the occurrence of HR in plant roots (Hermanns et al., 2003, Radwan et al., 2005). Additionally, previous studies have not shown any transcriptional parallels between common dPCD marker genes and HR PCD in Arabidopsis (Olvera-Carrillo et al., 2015; Salguero-Linares et al., 2022).

While *S. indica* is a beneficial root endophyte that does not induce classical hypersensitive response (HR) in host plants, the impact of dPCD on *S. indica* colonization should not be overlooked. *S. indica* promotes root hair formation in its hosts (Saleem et al., 2022), and in Arabidopsis, root hair cells naturally undergo cell death 2–3 weeks after emergence (Tan et al., 2016). This aspect could be particularly relevant for understanding the dynamics of *S. indica* colonization.

- Showing the analysis of pBFN1 in smb-3 would help in validating the idea that the downregulation of BFN1 by S. indica is regulated independently of SMB.

SMB is known to be a root cap specific transcription factor (Willemsen et al., 2008; Fendrych et al., 2014). The pBFN1:tdTOMATO reporter line shows that *BFN1* expression occurs in many different tissues undergoing dPCD, above and below ground, where SMB is not expressed or present. Therefore, we can postulate that the downregulation of *BFN1* by *S. indica* in the differentiation zone is regulated independently of SMB.

- A question of great interest still remains open: is it the microbe that induces the regulation of BFN1 causing a delay in cell clearance and favoring the infection or is it the plant that reduces BFN1 to favor the interaction with the microbe? In other words, is the mechanism a response to stress or a consolidation of the interaction with the host?

We agree with this reviewer that this question remains open. Whether active interference by fungal effector proteins, fungal-derived signaling molecules, or a systemic response of Arabidopsis roots underlies BFN1 downregulation during *S. indica* colonization remains to be investigated. Yet, it is noteworthy that the downregulation of *BFN1* in Arabidopsis is not specific to *S. indica* but also occurs during interactions with other beneficial microbes such as *S. vermifera* and two bacterial synthetic communities. This suggests that it could be a broader plant response to microbial presence. However, at this stage, we can only speculate on these possibilities. We therefore changed some of the statements in the paper to moderate our conclusions: e.g. “Expression of plant nuclease BFN1, which is associated with senescence, is modulated to facilitate root accommodation of beneficial microbes” to leave open who exactly is controlling *BFN1*, the plant or the microbes.

**Reviewer #2 (Recommendations For The Authors):**
This is a straightforward study, well executed and well written. I have only a few specific comments, and some concern the statistics which is a bit more serious and where I would like to get answers first. Looking at the figures, I am sure that the authors can easily clarify the issues in the manuscript.

We appreciate the positive feedback and included clarifications in the statistical section in the material and methods.

Statistics:- The statistics are not detailed in Material and Methods, but are only briefly indicated in the headings of the figures. Include a statistics section in Material and Methods.

We added an extra paragraph with statistical analysis in the Material and Method section for clarifications, which reads as follows:” All statistical analyses, except for the transcriptomic analysis, were performed using Prism8. Individual figures state the applied statistical methods, as well as p and F values. p-values and corresponding asterisks are defined as following, p<0.05 *, p<0.01**, p<0.001***.”.

- Figure 1/ Figure S3, etc: First of all, a **** with p< 0.00001 does not exist! Significance in statistics just means that we assume that there is a difference with some kind of probability that has been defined as p<0.05 *, p<0.01**, p<0.001***, and NOT more! Even if p<0.000001, it is still p<0.001***. Stating the meaning of asterisks in a separate Statistics section in Materials and Methods would also avoid repetitive explanations (e.g. Figure 4, L68: 'Asterisk indicates significantly different...').

We agree and have updated the manuscript accordingly. See comment above.

- Also, it is advisable to reduce the digits of the p-values to a meaningful length e.g. Figure 2 L 36: (*P<0.0466) should be (F[1, ?] = ?; p<0.047). The * is not necessary in the text, as p<0.05 is already given. We do not obtain more information by a more exact p-value, because all we need to know is that p<0.05.

We adjusted the p-values accordingly throughout the manuscript.

- It is NOT sufficient to communicate just the p-value of a statistical analysis. What is always needed is the F-value (student test and ANOVA) with both nominator and denominator degrees of freedom (e.g. F[2, 10]=) AND the p-value.

We included F-values throughout the manuscript in all main and supplemental figures to provide more clarity for the readers.

- The reason becomes clear in Fig. 2D where the authors state that they used 3 biological replicates, each with 40 plants. I assume the statistics was wrongly based on calculating with 120 plants (F[1,120]=) as technical replicates instead of correctly the biological replicates (3 means of 40 technical replicates each, (F[1,3]=))?? If F-value and df had been given, errors like this would be immediately visible - for any reviewer/reader, but also to the authors.=>Please re-analyze the statistics correctly.

To assess *S. indica*-induced growth promotion, we measured and compared the root length of Arabidopsis plants under *S. indica* colonization or mock conditions at three different time points. Each genotype and treatment combination involved measuring 50 plants, with each plant serving as an independent biological replicate inoculated with the same *S. indica* spore solution. For comprehensive statistical analysis, we conducted the experiment a total of 3 times, using fresh fungal inoculum each time, originally referred to as "three biological replicates." We maintain that including all plant measurements is essential for a thorough statistical analysis of our growth promotion experiment. However, in order to avoid confusion, we have updated the figure legend to clarify the experimental set-up as following: “(D) Root length measurements of WT plants and *smb-3* mutant plants, during *S. indica* colonization (seed inoculated) or mock treatment. 50 plants for each genotype and treatment combination were observed and individually measured over a time period of two weeks. WT roots show *S. indica*-induced growth promotion, while growth promotion of *smb-3* mutants was delayed and only observed at later stages of colonization. This experiment was repeater 2 more independent times, each time with fresh fungal material. Statistical analysis was performed via one-way ANOVA and Tukey’s post hoc test (F [11, 1785] = 1149; p < 0.001). For visual representation of statistical relevance each time point was additionally evaluated via one-way ANOVA and Tukey’s post hoc test at 8dpi (F [3, 593] = 69.24; p < 0.001), 10dpi (F [3, 596] = 47.59; p < 0.001) and 14dpi (F [3, 596] = 154.3; p < 0.001).”

- Figure 2, L 18; Figure 5, L 95, Figure S5 L53, etc: I am worried about executing a statistical test 'before normalization' - what does it mean?? WHY was a normalization necessary, WHAT EXACTLY was normalized and do we see normalized plots that do NOT correspond to the data on which the statistics was based? At least this implies 'before normalization'! Please explain, and/or re-analyze the statistics correctly.

We agree that the phrasing “before normalization” may lead to confusion, as the normalization of data to the mean of the control group does not alter the statistical analysis. Normalization was performed to achieve a clearer visual representation. Additionally, Evans blue staining is quantified by measuring the mean grey value, which does not correspond to a specific unit. Normalizing the data allows for the representation of relative staining intensities. The manuscript has been adjusted accordingly throughout.

- Statistics in Figure 1: L8/9: 'in reference to B' is unclear, I guess the mean of the control was used as a reference? This would also explain the variation in relative staining intensity (Figure 1C). if normalization was carried out (see above) all control (WT) values should be exactly 1, but they are not. I guess it was normalized to the mean of the control?

“In reference to X” or “corresponding to X” typically means that Figure X shows an example image from the dataset on which the statistical quantification is based. We have updated the manuscript throughout the main and supplemental figure legends to use “refers to image shown in X” to avoid confusion.

Figure S4, L 42: '(corresponding to A)', see comment above.

See comment above.

Figure 5B, L 87: '(in reference to A)'; L93: (in reference to C), etc. - see above. Unclear how A was used as a reference. Was it the mean of A? BUT again only 3 biological replicates! So it has to be the mean of 3 reps that was used as control! OR can we at least say that the 10 measured roots were independent of each other crucial (!) precondition for executing student's test or ANOVA? Then you would have at least 10 replicates (mean of 4 pictures taken per root for each).

Quantification of Evans blue staining intensity involved taking 4 pictures along the main root axis of each plant. We re-evaluated the statistical analysis correctly with the averaged datapoints for each plant root. We adjusted main figures (Fig.1C and 5B) and supplementary figures (Fig. S1C and S4B) and changed the material and methods section of the manuscript as following: “4 pictures were taken along the main root axis of each plant and averaged together, for an overview of cell death in the differentiation zone.”.

- Statistics in Figure 4, L 69: what means 'adjusted p-value'? Which analysis?

The material and method section of the manuscript has been adjusted as following for clarification: “Differential gene expression analysis was performed using the R package DESeq2 (Love et al., 2014). Genes with an FDR adjusted p-value < 0.05 were considered as differentially expressed genes (DEGs). The adjusted p-value refers to the transformation of the p-value obtained with the Wald test after considering multiple testing. To visualize gene expression, genes expression levels were normalized as Transcript Per kilobase million (TPM).”.

- Statistics in Figure 5, L102-105: see above! Were the statistics correctly calculated with 7 reps, or wrongly with 30? # I guess each time point was normalized to the mean of WT? By the way, it is not clear if repeated measurements were done on the same plants. If repeated measurements were done on the SAME plants, then these data are statistically not independent anymore (time-series analysis), and e.g. MANOVA must be used and significant (!) before proceeding to ANOVA and Tukey.

The statistics for quantifying intraradical colonization of Arabidopsis roots were calculated with 7 replicates. For each replicate, 30 plants were pooled to obtain sufficient material for RNA extraction and cDNA synthesis. Plants from the same genotype were harvested separately for each time point, ensuring that the time points are statistically independent from one another.

Statistics Fig. S1, L 11-12: see above, '5 plants were imaged for each mock and ..., evaluating 4 pictures ...' That means you have means of 4 pictures for 5 biological replicates - the figure shows 20 replicates. However, the statistics must be based on 5 reps! You may indicate the 4 pictures per root by different colours. Change throughout all figures and calculate the statistics correctly (show this by indicating the correct df in your statistics as discussed above).

We have conducted a re-evaluation of the statistical analysis of Evans blue staining for all figures presented throughout the manuscript. See comment above.

Statistics Fig. S3, L 31: 'Relative quantification of ...' see above, relative to what? Explain this also clearly in Statistics in Materials and Methods.

Relative quantification refers to normalizing data to the mean of the corresponding control group. Figure legends have been revised to clarify this point.

Statistics Fig. S5, L 57/58: 'Genes are clustered using spearmen correlation as distance measure'. If I understand it correctly, Spearman correlation is NOT a distance measure. You used Spearman correlation to cluster gene expression. Now it would be interesting to know WHICH clustering method was used, e.g. a hierarchical or non-hierarchical clustering method? and which one, e.g. single linkage, complete linkage? The outcome depends very much on the clustering method. Therefore, this information is important.

To perform gene clustering, we set the option “clustering_distance_rows = "spearman" “ of the Heatmap function included in the ComplexHeatmap package. The function first computes the distance matrix using the formula 1 - cor(x, y, method) with Spearman as correlation method. It then performs hierarchical clustering using the complete linkage method by default.

# Arabidopsis is a genus name and by convention, this has to be written throughout the MS in italics - even if the authors define *Arabidopsis thaliana* (in italics) = Arabidopsis (without).# typosL 24: smb-3 mutants (must be explained)L 83 insert: ...two well-characterized SMB loss-of-function ...

While *smb-3* is a SMB loss-of-function mutant *bfn1-1* is a BFN1 loss-of-function mutant, independent of SMB.

L 93: The switch between the biotrophic..L 119: distal borderL 125: aggregates in the smb-3 mutantL 132: between the meristematicL 177/178: was observed at 6 dpi in Arabidopsis colonized by S. indica.L 250: colonization stages by S. indica.L 288: and root cell death (RCD)L 289: and towards...L 296: dPCD protects theL 304: This raises theL 351: to remove loose

All the above-mentioned typos have been addressed in the manuscript.

Materials and MethodsL 327: give composition and supplier of MYP mediumL 344 name supplier of MS mediumL 338 name supplier of PNM mediumL 353: replace 'Following,..' with 'Subsequently, ..'L 360: replace 'on plate' with 'on the agar plate' - change throughout the Materials and methods!L 360: name supplier of Alexa Fluor 488L 363: name supplier of (MS) square plateL 377: insert comma: After cleaning, the roots...L 394: explain the acronym and name supplier of PBSL 399: explain the acronym and name supplier of TBST

All the above-mentioned comments in the material and methods have been addressed in the manuscript.

Figure 2G x-axis, change order: Hoechst/ProteostatFigure 3, L53: propidium iodide: name supplierFigure 4, L68: AsterisksL 60: explain LRCL 67, L69, L70: explain the acronym TPM and how expression values were measured in Materials and Methods, the brief explanation in the figure is unclear and not sufficient

All the above-mentioned comments in the figure legends have been addressed.

Figure S5, L50: explain 'SynComs'L 51: corrects 30 plans => 30 plantsL 56: vaules => valuesL 57: use capital letter: Spearman correlation

All the above-mentioned comments in the supplemental figure legends have been addressed.